# Examining the Correlates of HIV Testing for Venezuelan Migrants in Trinidad

**DOI:** 10.3390/ijerph20032148

**Published:** 2023-01-25

**Authors:** Nyla Lyons, Brendon Bhagwandeen, Jeffrey Edwards

**Affiliations:** 1Medical Research Foundation of Trinidad and Tobago, 7 Queen’s Part E, Port-of-Spain, Trinidad and Tobago; 2School of Mathematical and Computer Sciences, Heriot Watt University Malaysia, 1 Jalan Venna P5/2, Precinct 5, Putrajaya 62200, Malaysia; 3Department of Para-Clinical Sciences, Faculty of Medical Sciences, University of the West Indies, St. Augustine, Trinidad and Tobago

**Keywords:** government policy, HIV testing, migrants, Trinidad and Tobago

## Abstract

An important preventive measure in the fight against the HIV epidemic is the adoption of HIV testing. The government of the Republic of Trinidad and Tobago conducted a registration exercise in 2019 for undocumented migrants and refugees from Venezuela residing in the country. These migrants were allowed access to the public health system. In this study, we observed the correlates of HIV testing in Venezuelan migrants residing in Trinidad. A convenience sample of *n* = 250 migrants was collected via telephone survey from September through December 2020. Variables of interest included social factors, health needs, and uptake of HIV testing. Pearson χ^2^ tests examined the associations between study variables, and multivariable logistic regression with backward elimination produced the odds of taking an HIV test. In our study, 40.8% of migrants reported having received an HIV test since arriving in Trinidad. Persons who migrated with family or friends had greater odds of getting an HIV test relative to persons who arrived alone (OR = 2.912, 95% CI: 1.002–8.466), and migrants who knew where to get an HIV test also greater odds of getting a test relative to person who did not know where to get a test (OR = 3.173, 95% CI: 1.683–5.982). Migrants with known physical health problems had greater odds of getting an HIV test relative to migrants without these health problems (OR = 1.856, 95% CI: 1.032–3.337). Persons who arrived with family or friends had greater odds of experiencing difficulties accessing public health care relative to persons who arrived alone (OR = 3.572, 95% CI: 1.352–9.442). Migrants earning between $1000 and $2999 TT per month had greater odds of experiencing trouble accessing public health services relative to persons who had monthly earnings of less than $1000 TT (OR = 2.567, 95% CI: 1.252–5.264). This was the first quantitative study on HIV testing among Venezuelan migrants in Trinidad. Migrants still experience difficulties accessing healthcare, which, in turn influences national HIV prevention and control efforts. The results gathered may help in developing HIV prevention plans that are led by a national health policy that takes migrant communities’ needs into account.

## 1. Introduction

Beginning in 2016, Trinidad and Tobago experienced increasing inflows of refugees and migrants from Venezuela. By the end of 2017, the United Nations High Commissioner for Refugees (UNHCR) reported a total of 2700 refugees seeking asylum, and by April 2018, the number of reported refugees increased to over 5300 [1]. In December 2019, the UNHCR estimated 23,400 refugees and migrants from Venezuela were residing in Trinidad and Tobago, and this was expected to increase to 33,400 by the end of 2020 [2]. In June 2019, refugees, and migrants from Venezuela (including undocumented persons) were given a two-week amnesty and an opportunity to obtain one-year legal status through a government registration exercise. Successful registrants were granted a one-year work permit and benefited from free emergency medical and primary care services available in the public health system [3,4].

The health sector of Trinidad and Tobago is comprised of public and private care systems. The Ministry of Health (MOH) is responsible for policies, planning and coordination at the national level. At the sub-national level, there are currently five (5) Regional Health Authorities (RHAs), each responsible for the delivery of primary and secondary health services specifically in the north, west, central, south, and east regions of Trinidad and Tobago. Apart from the RHAs, the MOH oversees HIV surveillance, the provision and treatment of sexually transmitted infections with designated facilities. Standards, protocols, and guidelines for the provision of basic health care are the responsibilities of the MOH and a referral system facilitated the integration and coordination of the different levels of care [5].

Access to healthcare in the public system is free and financed through the government and taxpayers. Venezuelans and other non-nationals may access primary and secondary care services on a walk-in basis through publicly funded hospitals and smaller health centers or clinics [6]. There are six (6) public hospitals located throughout the country and smaller health centers in each of the five RHAs which conduct HIV testing free of charge. In the private health system, persons are expected to pay up front when seeking services such as HIV testing at a privately funded laboratory. The population can also access HIV testing through community interventions and through non-governmental organizations (NGOs) which conduct outreach to vulnerable populations which are most at risk for HIV infection [7]. In 2010, the government expanded the availability of same-day HIV testing and mobile outreach programs to increase the uptake of and access to HIV testing services [6].

In 2014, the UNAIDS announced its ambitious target aiming to end the AIDS epidemic by 2030 by achieving a 95% diagnosed status among all people living with HIV (PLHIV), 95% on antiretroviral therapy (ART) among diagnosed, and 95% virally suppressed (VS) among those treated [8]. Therefore, it is critical for countries to increase coverage of HIV testing, prevention, and treatment to fulfil their commitment to achieve the 95-95-95 global targets. Preventing new HIV infections and other STIs among general, key, and vulnerable populations is one of the key objectives outlined in the Trinidad and Tobago National HIV and AIDS policy. This included expanding HIV testing coverage among young people, women, men who have sex with men (MSM), sex workers, people who use drugs, migrants and other vulnerable populations [8,9].

It was well documented that the conditions of migration and the lack of appropriate policy responses by countries may exacerbate health risks and increase vulnerability for migrant populations [10]. This included their living and working conditions and particularly their migration status which may leave them more vulnerable to health risks and less able to cope with illness, including HIV-related illness [11,12].

The availability of healthcare services for Venezuelan refugees and migrants is a growing concern in Trinidad and Tobago. Currently, there is no national policy consistent with international law and standards governing their access to health services. Migrants in need of international protection remain subjected to the provisions of the 1976 Immigration Act. The government acceded to the 1951 Geneva Convention on the Status of Refugees, and its 1967 Protocol in November 2000. In 2014, a draft National Policy was developed to address refugees and asylum matters and is yet to be implemented [13].

Given the increasing inflows of Venezuelan migrants to Trinidad and Tobago, it is important to understand the factors affecting the uptake of HIV testing and its implications for the public health system. This is especially important in the absence of health policy to address the primary care and preventative health needs of migrants and their families.

Therefore, the objectives of this study were to (1) describe the uptake of HIV testing among Venezuelan migrants and (2) examine the correlates and predictors of HIV testing and their access to health services.

## 2. Methods and Methods

### 2.1. Study Design and Participants

This cross-sectional study was carried out from September to December 2020 among Venezuelan migrants living with HIV in Trinidad, by telephone interviews using a structured questionnaire due to the COVID-19 pandemic. The interviews were conducted by trained bilingual nurses at the Medical Research Foundation of Trinidad and Tobago (MRFTT). The MRFTT is the largest HIV Treatment and Care Centre in Trinidad and Tobago, where daily clinics are held via appointments and walk-in visits.

The study employed two sampling strategies. Given that the Venezuelan migrants comprised a hidden population in Trinidad, a convenience sample was generated based on migrants responding to flyers with information about the study, that were distributed to a large community-based organization which provided refugee support services to Venezuelan migrants. Firstly, contact was initiated by migrants who were interested in enrolling in the study and subsequent free health check-ups. After which, the migrants would share details of the study for snowball sampling to be employed to recruit further participants. In total, responses came from a sample of *n* = 250 Venezuelan migrants residing in Trinidad who met the eligibility criteria and were willing to participate.

The eligibility criteria included persons of Venezuelan nationality living in Trinidad between 3 months and 5 years. Once the criteria were deemed eligible by the nurses, these persons received information about the study protocol and participated after their consent to participate was provided. Participants were informed that they could stop completing the interview at any time. All responses were translated and recorded in English by the nurses on an Excel spreadsheet stored on a password protected computer. Only members of the research team had access to the data file.

### 2.2. Ethical Approval

Permission to conduct this study was granted by the University of the West Indies (UWI) Institutional Review Board (IRB) in September 2020.

### 2.3. Study Questionnaire and Data Collection

In this study, sociological factors included age, gender, time since arrival in Trinidad, having migrated alone or with family and living alone or with family. Other factors explained barriers and facilitators to health service use, including educational level, employment status, self-rated English fluency, access to information on health services (HIV testing), where to access health services (HIV testing), overall self-rated health condition, being worried about a health condition and whether a doctor was visited in the past 12 months. The outcomes variables focused on the uptake of HIV testing by migrants and whether they had difficulty in accessing health services.

A health questionnaire was developed by health research professionals at the MRFTT and translated into Spanish. The study questionnaire was pre-tested using a convenience sample of persons randomly selected from a health database to help identify potential problems in the language, structure, and design of the questionnaire. Some minor adjustments were subsequently made to the questionnaire to better facilitate the flow of interview process. Each interview took approximately 15 min to complete.

### 2.4. Other Variables

Additional socio-demographic variables such as income level and reason for migration; and health/health service variables including reason for accessing healthcare providers and knowledge of where to access health services (HIV testing, pregnancy testing, condoms) were included in the context of this study.

### 2.5. Statistical Analysis

The Statistical Package for the Social Sciences (SPSS) version 22.0 was used for statistical analysis. Descriptive statistics (n (%) for categorical variables) were used to characterize baseline distributions of study variables. Univariate associations between socio-demographic and health/health service variables with having received HIV testing and having difficulty in accessing health services were observed using Pearson χ^2^ tests. Explanatory logistic regression with backward elimination was used to examine adjusted odds ratios (OR) (with 95% confidence intervals (CI)) between outcome variables with social factors and health needs, controlling for potential confounders. *p* ≤ 0.05 was considered statistically significant.

## 3. Results

### Socio-Demographic Characteristics and Health Needs of Participants

A total of 250 persons completed interviews in this study. There were more females (*n* = 149, 59.6%) than males (*n* = 101, 40.4%), with most of the sample (*n* = 221, 88.4%) reporting some form of education. Twenty-two (8.8%) of migrants arrived in Trinidad alone, while 228 (91.2%) arrived with family/friends. Almost half of the participants (*n* = 131, 52.4%) indicated they were employed, while 145 (58%) reported earning an income of less than $1000 TT. The most prevalent age group was 30–39 years (*n* = 93, 37.2%). Thirty-seven of the participants (14.8%) reported that their length of stay in Trinidad was 12 months or less. When asked about their level of English fluency, 157 (62.8%) of the study participants reported that they were not able to speak English well, 81 (32.4%) reported moderate fluency in English and 12 (4.8%) reported that they were able to speak English reasonably well. Most of the migrants (*n* = 227, 90.8%) indicated that they migrated to Trinidad for economic reasons. Table 1 summarized the socio-demographic characteristics of the participants.

Twenty-four (9.6%) persons perceived themselves as “not very healthy”, 74 (29.6%) reported being in “fair health” and 152 (60.8%) persons reported being in “good health”. When asked, 70 (28%) indicated they had a physical health problem (including diabetes, hypertension, sexual/reproductive health problems, kidney, heart, or other chronic diseases) that they were worried about, and 79 (31.6%) reported being worried about having a mental health problem (including anxiety, stress, difficulty sleeping, difficulty concentrating or any other condition affecting mental health). Almost half of the participants (*n* = 123, 49.2%) indicated they would most likely go to a public health center to get an HIV test, 28 (11.2%) indicated they would go to a community center or NGO and 27 (10.8%) persons indicated that they would get an HIV test through another health facility if necessary. Of the total study sample, 178 (71.2%) persons reported knowing where to get an HIV test. To get a pregnancy test, 118 (47.2%) participants indicated that they would go to a public health clinic, 8 (3.2%) indicated they would go to a community health center/NGO, 82 (32.8%) persons indicated they would go to another health event or private healthcare provider, and 42 (16.8%) indicated that they were unsure of where to get a pregnancy test if needed. For persons wanting to obtain condoms, 50 (20%) indicated that they would get them from public health centers, 8 (3.2%) indicated they would get condoms if they went to a community health event/NGO, 148 (59.2%) indicated they would get condoms from another health facility, and the remaining 46 (18.4%) indicated that they were not sure of where to go to get condoms if needed. Out of the total study sample, approximately two-thirds (*n* = 167, 66.8%) of participants reported having trouble while using the public health services in Trinidad, while 102 (40.8%) indicated that they had an HIV test since arriving in Trinidad. Table 2 summarized the health service characteristics of the participants.

Table 3 and Table 4 showed the results of the χ^2^ tests of association and multivariable logistic regression for migrants receiving an HIV test since residing Trinidad and Tobago. It was observed that receiving an HIV test had statistically significant associations with the independent factors, migrants having a known physical health problem about which they were concerned (*p* = 0.033), knowledge of the availability of HIV testing (*p* < 0.001), and the most likely facility migrants would go to for an HIV test (*p* = 0.002).

The factors from our univariate analysis were initially included in the multivariable model. Given backward elimination was used to choose the variables which best explained whether migrants received an HIV test while residing in Trinidad. The odds of persons who arrived in Trinidad with family or friends receiving an HIV test were almost three times the odds of persons who arrived alone (OR = 2.912, 95% CI: 1.002–8.466). The odds of migrants with known physical health problems receiving an HIV test were almost twice the odds of migrants without these health problems (OR = 1.856, 95% CI: 1.032–3.337). The odds of persons with knowledge of the availability of HIV testing receiving an HIV test were over three times the odds of persons who were not aware of where testing was available (OR = 3.173, 95% CI: 1.683–5.982).

Table 5 and Table 6 showed the results of the χ^2^ tests of association and multivariable logistic regression for migrants having trouble accessing public health services while in Trinidad. It was observed that having trouble accessing public health services had statistically significant associations with the migrants’ main reason for migrating to Trinidad (*p* = 0.016), with whom the migrants arrived in Trinidad (*p* = 0.007), monthly earnings (*p* = 0.032), time since arrival in Trinidad (*p* = 0.031), knowledge about the availability of HIV testing services (*p* = 0.035), most likely place migrants would go for HIV testing (*p* = 0.008), most likely place migrants would go for a pregnancy test (*p* = 0.050), and most likely place migrants would go to get condoms (*p* = 0.020).

All factors from our univariate analysis were initially included in the multivariable model. Backward elimination was used to choose the variables which best explained whether migrants had difficulties accessing public health services in Trinidad. The odds of migrants who arrived in Trinidad with family or friends having trouble accessing public health services were more than three times the odds of migrants who arrived alone (OR = 3.572, 95% CI: 1.352–9.442). The odds of persons earning between $1000 and $2999 TT per month having trouble accessing public health services were over twice the odds of persons who had monthly earnings of less than $1000 TT (OR = 2.567, 95% CI: 1.252–5.264). The odds of persons using private health care facilities or health events to get condoms and having trouble accessing public health services were lower than the odds of persons who used the public health facilities (OR = 0.264, 95% CI: 0.110–0.633).

## 4. Discussion

Most published studies on HIV testing and migrants were conducted in Europe and North America [14]. To our knowledge, this was the first quantitative study among Venezuelan immigrant community in Trinidad and the English-speaking Caribbean. Our study aimed to examine the uptake and factors associated uptake of HIV testing among Venezuelan migrants in Trinidad.

We found a 40.8% prevalence of having been tested for HIV among study participants. Few studies have examined the reasons for HIV testing among migrants. Those that have indicated that migrants often seek HIV testing for prenatal care, their perceived risk of contracting HIV and/or other STIs, general health check-ups, and health problems such as the need for surgery [15,16,17]. Although our study did not examine the reasons for HIV testing, we found that reasons for accessing a medical health provider, knowledge of where to access free HIV testing and facility through which they would get an HIV test were associated with HIV testing in Venezuelan migrants in Trinidad. When asked about the availability of HIV testing, almost 71% of the participants in our study indicated that they received information through their social networks and the public healthcare system. Similar findings were found in studies examining the relationships between social networks and the uptake of HIV testing among Latino immigrants, and in a large study of HIV risks behaviors among male migrants in China. Both studies concluded that social network factors and core network membership were important factors in understanding HIV transmission and designing risk-reduction interventions in the migrant population [18,19]. The results also underscored the relationship between migrant’s use of social networks to obtain HIV information and using it. Providing information on how and where to access HIV health services may help to ensure adequate and timely use of these services. Consequently, it may contribute to improving uptake of HIV testing and early HIV diagnosis. It also demonstrated the need to improve information on the availability of HIV testing and promote the use of health services through migrants and their social networks.

One of the primary goals of National Program for HIV/AIDS in Trinidad and Tobago has been to expand access to HIV testing to promote early detection of HIV infection [20]. As a result, efforts have been undertaken to ensure routine voluntary testing was available in the public health system. This has been offered free of charge, in co-operation with non-governmental organizations and through community events to promote HIV testing among vulnerable groups, including migrants. Community-driven HIV testing increased access to services and appropriate referral for vulnerable populations such migrants who may be stigmatized and found it difficult to access health care services available in the public health system [6,21]. In our sample, most of the participants indicated they would use the public health system if they wanted an HIV test. However, the study also found that 28.8% of persons indicated they did not know where to go to get a HIV test or did not receive any information on where they could get an HIV test. These finding were consistent with the existing literature which showed that a lack of awareness of the availability of health services may serve as a barrier to service use among migrants [22]. The results also demonstrated the gaps in national public health efforts and HIV education campaigns to reach vulnerable populations, including migrants.

As it pertains to related sexual services, 32.8% of the study participants indicated that they would go a private healthcare provider such as a pharmacy or a private doctor to get a pregnancy test if they needed it, as compared to 47.2% who indicated they would use public health centers if they needed a pregnancy test. Furthermore, almost 60% of persons indicated that would also go to a private healthcare provider if they needed condoms while 20% indicated that they would use a public health center to access condoms. These finding also highlighted the gaps in access to sexual health services, and/or unavailability of information on the availability of these services in the public health sector which translated into missed opportunities for HIV prevention.

In our study, reasons to access medical health providers as measured by having seen a doctor (since in Trinidad for a physical health problem) were independently associated with receiving an HIV test but not a significant determinant of HIV testing in the presence of other factors. The association showed the potential opportunities to integrate HIV prevention within the health system, also as an opportunity to improve early diagnosis of HIV infection.

Our study also looked at the factors associated with the difficulties using public health services (including services for HIV testing) as reported by Venezuelan migrants. Existing literature highlighted the influence of factors at individual, provider, institutional and policy levels that were deterrents to HIV testing among migrants and their use of health services [23]. Studies have also cited the influence of financial problems, language and culture as barriers to migrant’s access to the healthcare [9,20].

On the individual level, our study found that the most likely facility for migrants to get HIV tests, pregnancy tests and condoms were independently associated with difficulties while using public health services. This may be attributed to inadequacy of public health system to address the health needs of Venezuelan migrants accessing these services given that at most half of the migrants would use this facility. Our study showed that migrants who arrived with family/friends had greater odds of reporting difficulties using public health services compared to those who arrived alone. Additionally, monthly earnings were independently associated with reporting difficulties in accessing public health services. These differences may be explained by the fact that younger migrant women earned less, were mostly of reproductive age, and more likely to use health services related to sexual and reproductive health, such as prenatal care. Many migrants in the study reported using the private healthcare provider for pregnancy testing and to obtain condoms. While in Trinidad, universal antenatal care, including HIV testing, was available, little has been published on difficulties faced by Venezuelan women in accessing antenatal services. Several studies have concluded that undocumented migrants were more likely to report less utilization of health services [24,25]. There was also evidence showing that the underuse of health services hindered the uptake of HIV testing [26]. Our study did not measure the difference in women’s access to HIV testing based on their legal status (i.e., documented, or undocumented).

Migrant women may also face barriers to accessing HIV health services due to mistrust of health systems, confidentiality, fear of discrimination and exclusion, and lack of awareness of the services available may be reasons for not having been tested, as pointed out by other authors [27]. In this context, innovative strategies to promote and increase the use of health services, including publicly available HIV testing should be developed, with a focus on addressing the sexual health needs of migrant women.

Our study did not show an association between fluency in speaking English and having trouble while using health services. However, linguistic differences have been pointed out in several studies as a factor associated with underutilization of sexual health services among immigrants [19]. In Trinidad, HIV testing was performed by trained health care providers and lay testers, many of which possessed basic fluency in speaking Spanish.

Previous studies have shown that understanding the health system was associated with higher levels of HIV knowledge, awareness of availability of health services which translated into higher rates of HIV testing [28]. Persons who obtained information on HIV testing through social networks were more likely to report difficulties which using the public services. Our results further demonstrated the need to improve the cultural competency of public healthcare system (and health staff) to better assist migrants to navigate the health system, however additional studies would be needed to better understand these relationships.

The adoption of migrant health policies that were responsive to the needs of migrants may help address barriers to accessing health services in host countries, therefore, contributing to, and promoting earlier HIV diagnosis. The results of two large studies conducted in Europe and Africa found that migrants were not able to access the health services they needed due to several structural-level factors to include organizational and policy-related determinants [29]. In Trinidad, while there were no requirements for persons, regardless of nationality, to access publicly available health services including HIV testing free of charge, there was no clear health policy to address the ongoing care of Venezuelan migrants.

In March 2019, in response to the increasing inflows of Venezuelan migrants to Trinidad and Tobago, the government conducted a registration exercise granting one-year access to primary care and emergency medical services for just over 16,000 documented and undocumented Venezuelan migrants at the time. There was an attempt for a second registration exercise in March 2021. The sum of efforts so far has focused more on managing borders to limit entry to Venezuelan migrants rather than reducing the barriers to accessing to health services they needed. Notwithstanding the one-time registration exercise, migrants from Venezuela continued to face challenges in accessing health services they needed which impacted prevention. Testing coverage continued to pose challenges.

There is no immigrant registration system in place making it difficult to accurately assess the inflows and outflows of Venezuelan migrants over a given period. In this regard, the population size of Venezuelan migrants is unknown and estimated solely through the UN Agency reports on registered refugees, and asylum seekers. Apart from this, no population size estimates, or national-level surveys have been conducted on migrant populations. Our survey was best able to capture a cross-section of Venezuelan migrants seeking health services over the course of the three-month period, and given the transient nature of the population, the survey results presented a reasonable representation of their health seeking behaviors and needs, especially in the absence of population-based samples.

The ability to reach the Venezuelan migrant population through surveys, and more specifically to understand their uptake of health services may also be affected by stigma. The Venezuelan migrant community faces stigma in general which creates barriers affecting their access to, and uptake of health services both in the public and private sectors. These barriers may be due to financial, language and cultural factors which occur because of the healthcare system itself and/or because of health care providers, and ultimately impacts the ability to reach a desired sample size. One of the strengths of our study was that it was administered through a non-governmental agency—the Medical Research Foundation—which itself served to reduce stigma and increase the likelihood of participation from the community of Venezuelan migrants. Additionally, our survey questionnaire was administered by bilingual Venezuelan nurses who are culturally competent, further increasing the successful uptake and acceptance of the survey by the study population. While the study population was reached primarily via telephone, persons expressing immediate health needs were provided with follow-up case management and access to free medical screening, also serving to minimize any financial barriers. Based on what we know now, lessons can be gleaned for health authorities at national, subnational, and local levels, as well as health care practitioners on successful approaches and techniques to promote migrant participation in surveys with further implications for their participation in policy planning and practice.

## 5. Study Limitations

One of the biggest limitations of this study was the sample size of 250 observations. There are thousands of undocumented migrants from Venezuela in Trinidad and Tobago as in the case of the Government Registration Exercise where over 16,000 migrants participated. Therefore, the statistical inferences based on this sample should not generalized to the larger population of Venezuelans living in Trinidad.

Another, limitation was that some of these findings may be biased due in part to the selection of study participants. Some the migrants elicited for this study were selected from a database of persons who previously accessed outreach health services and who were seeking refugee status determination (i.e., were documented). However, despite these limitations, our study was vital as it examined the factors associated with the uptake of HIV testing among a highly vulnerable population and, to our knowledge, has not been studied before in Trinidad and the wider English-speaking Caribbean. The response rate to the study was high, and we are confident that the study yielded value as migrants have been regarded as a hidden population given the high level of stigma, many of whom did not want to access public health services because of their legal status. Our study did not assess the role of stigma; however, migrants may experience difficulties while using public health services due to HIV-related societal and/or internalized stigma.

## 6. Conclusions

A greater understanding of the factors associated with HIV testing among the Venezuelan migrant population was relevant for the design of tailored health messages to best address their needs. Future studies should also examine the association of HIV stigma especially as it relates to their use of public health services and access to HIV testing. This information would be valuable to public health officials in developing initiatives focused on promoting HIV testing and effective prevention strategies to include the dissemination of information about the availability of HIV health services and promoting its use. This was critical as migrants tend to not be reached by prevention and treatment services. Effective strategies may include promoting participation of migrant communities and their networks in the planning and development of HIV prevention interventions, including community-based voluntary testing.

These strategies and interventions should be guided by health policy that would be inclusive, specifically addressing the health needs and expanding access to primary care services to include HIV testing, counselling, and linkage to treatment. The study underscored the value of migrant networks and NGOs which was complementary to any strategy to reduce stigma and expand reach to migrant communities. An effective Information, Education and Communication (IEC) health prevention and education campaign aimed at raising levels of knowledge about HIV, about where to access health services and targeting the needs of sub-populations may help reduce barriers to the use of health services.

## Figures and Tables

**Table 1 ijerph-20-02148-t001:** Socio-demographic characteristics of participants.

Variables	*n*	%
Gender		
Female	149	59.6
Male	101	40.4
Total	250	100.0
Education		
No formal education	29	11.6
Primary	94	37.6
Secondary	120	48.0
Tertiary	7	2.8
Total	250	100.0
Age		
20–29 years	72	28.8
30–39 years	93	37.2
40–49 years	59	23.6
>50 years	26	10.4
Total	250	100.0
Reasons for migration		
Family reunification	12	4.8
Economic reasons	227	90.8
Escape from violence/asylum seeker	12	4.8
Total	250	100.0
With whom did you arrive in Trinidad?		
Arrived alone	22	8.8
Arrived with family/friends	228	91.2
Total	250	100.0
English fluency		
Low	157	62.8
Moderate	81	32.4
High	12	4.8
Total	245	100.0
Employment status		
Unemployed	119	47.6
Employed	131	52.4
Total	250	100.0
Monthly earnings		
<$1000 TT	145	58.0
$1000–$2999 TT	70	28.0
>$3000 TT	35	14.0
Total	250	100.0
Time since arrival in Trinidad		
≤12 months	37	14.8
>12 months	213	85.2
Total	250	100.0

**Table 2 ijerph-20-02148-t002:** Health needs and services of participants.

Variables	*n*	%
Self-rated health		
Poor health	24	9.6
Fair health	74	29.6
Good health	152	60.8
Total	250	100.0
Known mental health problem		
No	171	68.4
Yes	79	31.6
Total	250	100.0
Known physical health problems		
No	180	72.0
Yes	70	28.0
Total	250	100.0
Knowledge of where to get an HIV test in Trinidad		
No	72	28.8
Yes	178	71.2
Total	250	100.0
Where would you most likely go to get an HIV test?		
Public healthcare providers	123	49.2
Community center/NGO	28	11.2
Other	27	10.8
Unsure	72	28.8
Total	250	100.0
Where would you most likely go to get a pregnancy test?		
Public healthcare providers	118	47.2
Community center/NGO	8	3.2
Other	82	32.8
Unsure	42	16.8
Total	250	100.0
Where would most likely you go to get condoms?		
Public healthcare providers	50	20.0
Community center/NGO	6	2.4
Other	148	59.2
Unsure	46	18.4
Total	250	100.0
Had an HIV test since arriving in Trinidad?		
No	148	59.2
Yes	102	40.8
Total	250	100.0
Had trouble while using public health services?		
No	83	33.2
Yes	167	66.8
Total	250	100.0

**Table 3 ijerph-20-02148-t003:** Factors associated with having received an HIV test while in Trinidad.

Factors	*p*-Value
Sociodemographic	
Gender	0.563
Education level	0.491
Age	0.141
Main reason for migration	0.160
With whom migrant arrived	0.071
English fluency	0.082
Employment status	0.153
Monthly earnings	0.105
Time since arrival	0.743
Health needs and services	
Self-rated health	0.478
Known mental health problems	0.537
Known physical health problems	0.033
Knowledge of where to get an HIV test	<0.001
Most likely place to get an HIV test	0.002
Most likely place to get a pregnancy test	0.094
Most likely place to get condoms	0.961
Trouble using public health services	0.610

**Table 4 ijerph-20-02148-t004:** Correlates of having received an HIV test while in Trinidad.

Predictors	*p*-Value	OR	95% CI
Arrival with whom			
Arrived alone	-	Ref	-
Arrived with family/friends	0.050	2.912	1.002–8.466
English fluency			
Low	-	Ref	-
Moderate	0.238	0.706	0.396–1.259
High	0.051	0.207	0.043–1.005
Known physical health problems			
No	-	Ref	-
Yes	0.039	1.856	1.032–3.337
Knowledge of where to get an HIV test			
No	-	Ref	-
Yes	<0.001	3.173	1.683–5.982

Ref: Reference category.

**Table 5 ijerph-20-02148-t005:** Factors associated with having trouble accessing public health services while in Trinidad.

Factors	*p*-Value
Sociodemographic	
Gender	0.688
Education level	0.182
Age	0.469
Main reason for migration	0.016
With whom migrant arrived	0.007
English fluency	0.737
Employment status	0.081
Monthly earnings	0.032
Time since arrival	0.031
Health needs and services	
Self-rated health	0.558
Known mental health problems	0.824
Known physical health problems	0.943
Knowledge of where to get an HIV test	0.035
Most likely place to get an HIV test	0.008
Most likely place to get a pregnancy test	0.050
Most likely place to get condoms	0.020

**Table 6 ijerph-20-02148-t006:** Correlates of having trouble accessing public health services while in Trinidad.

Predictors	*p*-Value	OR	95% CI
Main reason for migration			
Economic reasons	-	Ref	-
Escape from violence/asylum seeker	0.298	0.487	0.126–1.886
Family reunification	-	-	-
Arrival with whom			
Arrived alone	-	Ref	-
Arrived with family/friends	0.010	3.572	1.352–9.442
Monthly earnings			
<$1000 TT	-	Ref	-
$1000-$2999 TT	0.010	2.567	1.252–5.264
>$3000 TT	0.310	1.567	0.659–3.727
Time since arrival			
≤12 months	-	Ref	-
>12 months	0.060	2.128	0.967–4.681
Knowledge of where to get an HIV test			
No	-	Ref	-
Yes	0.071	1.845	0.948–3.591
Most likely place to get condoms			
Public health facility	-	Ref	-
Community centre/NGO	0.570	0.572	0.083–3.949
Other	0.003	0.264	0.110–0.633
Unsure	0.197	0.483	0.160–1.460

Ref: Reference category.

## Data Availability

The raw data supporting the conclusions of this article will be made available by the authors if requested, without undue reservation.

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
