# Peer review of "Examining the Correlates of HIV Testing for Venezuelan Migrants in Trinidad"

_ijerph, 2023, doi:10.3390/ijerph20032148_

Round 1
Reviewer 1 Report
This is well written paper about an interesting and timely research project. The data gathering process and the statistical methods used for the analysis are appropriate. Moreover, the interpretation of the results stated in the Conclusions section is justified. Hence my recommendations are more a matter of style than about the substance of the paper.
My main recommendation is the following: As the authors mention the sample size and selection process of the participants are the main limitations of their project. Although it is commendable to recognize these limitations and I agree with the authors that, despite its constrains, this study is vital given its study subject, I still would like to know their impressions about whether the sample collected by them is representative or not of the population that they are trying to study. This can be based only on their expertise in the area or what has happened in previous similar cases in the literature. They mention that “Future studies should also examine the association of HIV stigma especially as it relates to their use of public health services and access to HIV testing” but in the meantime it would be interesting to know their informed opinion about the magnitude of the effect of this stigma and other factors that might have biased the sample, e.g. requirement to have access to telephone, based on what it is known now.
Other minor suggestions are:
· - In the Abstract the term “Univariate analyses” (line 18) is mentioned but given that the only univariate analysis used is the Pearson X^2 tests, it might better to change the current general term for the specific test that was selected.
· - In the Abstract it is mentioned that the OR’s come from a logistic regression model but not the method used to find the p-values.
· -In the Abstract instead of the variable Gender (line 33), please mention which specific gender has more trouble using public health services.
· - Clarify how the 16,00 persons mentioned in line 360 are related to the 33,400 listed in line 40. Are they a subset? Are they a different set of persons?
· - Lines 169 to 197 are basically just a description of Table 2 and hence this text is redundant. Please find a way to reduce this content and just highlight the most important results.
· - Table 2 can be extended to accommodate the p-values mentioned in lines 150-153. In that way Tables 3 and 5 become unnecessary. This will result in a better use of the space dedicated to the paper.
Reviewer 2 Report
In the paper by Nyla Lyons, Brendon Bhagwandeen, and Jeffrey Edwards, the authors assess the associations between different demographic and sociological factors and HIV testing in a small group of Venezuelan migrants in Trinidad.
My main concern for this study is the representativeness of the sample. First, the sample is small, 250 individuals. Even assuming random sampling (which is not the case), this sample size would reach a very low precision for the different estimates. This should be reflected by reporting the point estimates and the 95% confidence intervals.
The second issue is the sampling design, which is prone to participation bias. This should be addressed with some sort of correction (see, for example, Gray, Linsay, et al. "Correcting for non-participation bias in health surveys using record-linkage, synthetic observations and pattern mixture modeling." Statistical methods in medical research 29.4 (2020): 1212-1226.) or acknowledging the inherent limitations of the sampling design in this study regarding valid statistical inference.
Another critical issue is the statistical analysis methodology. It is not statistically sound to perform multivariable regression by including only the significant predictors from univariable analyses as covariates (for example, see: Sun, Guo-Wen, Thomas L. Shook, and Gregory L. Kay. "Inappropriate use of bivariable analysis to screen risk factors for use in multivariable analysis." Journal of clinical epidemiology 49.8 (1996): 907-916.).
Regarding the results, it is worrisome that some of the questions have such a high non-response rate: 23.2% for the question "With whom did you arrive?" and 47.2% for the question "Self-rated health". This is especially important for these two questions since they are significant in the chi-squared tests and were therefore included in the multivariable logistic regression models for HIV testing and having problems accessing public health services. How were these variable included in the logistic regressions? Is "no response" a category? Or were all the participants with "no response" removed from the analyses?
Finally, not only the statistically significant results should be presented. All results from the different analyses should be reported in the paper, independently of their p-value.
Round 2
Reviewer 2 Report
The authors have addressed all my concerns, and the manuscript has significantly improved. Nevertheless, there are still some minor issues that need attention:
The interpretation of odds ratio is not correct (line 21). An odds ratio of 3.52 does not mean that "persons who migrated with family or friends were over three times more likely to have an HIV test relative to persons who arrived alone." An odds ratio is not a relative risk. Please amend the explanation. The same applies to the other interpretations of odds ratios throughout the manuscript.
Following my suggestion, the authors modified the original modeling procedure, which was based on including only statistically significant univariate factors. They decided to include all the predictors in the multivariable analysis. This is fine, but taking into account the large number of covariables and the limited sample size, the authors should consider, at least, some sort of penalized mode (for example ridge regression) as a sensitivity analysis.
